

# Rapidly increasing macroalgal cover not related to herbivorous fishes on Mesoamerican reefs

Adam Suchley[1,2], Melanie D. McField[3] and Lorenzo Alvarez-Filip[2]

[1] Posgrado en Ciencias del Mar y Limnología, Instituto de Ciencias del Mar y Limnología, Universidad Nacional Autónoma de México, Ciudad de México, México
[2] Unidad Académica de Sistemas Arrecifales, Instituto de Ciencias del Mar y Limnología, Universidad Nacional Autónoma de México, Puerto Morelos, Quintana Roo, México
[3] Healthy Reefs for Healthy People Initiative, Smithsonian Institution, Ft Lauderdale, Florida, USA

Corresponding author
Lorenzo Alvarez-Filip,
lorenzoaf@gmail.com

## ABSTRACT

Long-term phase shifts from coral to macroalgal dominated reef systems are well documented in the Caribbean. Although the impact of coral diseases, climate change and other factors is acknowledged, major herbivore loss through disease and overfishing is often assigned a primary role. However, direct evidence for the link between herbivore abundance, macroalgal and coral cover is sparse, particularly over broad spatial scales. In this study we use a database of coral reef surveys performed at 85 sites along the Mesoamerican Reef of Mexico, Belize, Guatemala and Honduras, to examine potential ecological links by tracking site trajectories over the period 2005–2014. Despite the long-term reduction of herbivory capacity reported across the Caribbean, the Mesoamerican Reef region displayed relatively low macroalgal cover at the onset of the study. Subsequently, increasing fleshy macroalgal cover was pervasive. Herbivorous fish populations were not responsible for this trend as fleshy macroalgal cover change was not correlated with initial herbivorous fish biomass or change, and the majority of sites experienced increases in macroalgae browser biomass. This contrasts the coral reef top-down herbivore control paradigm and suggests the role of external factors in making environmental conditions more favourable for algae. Increasing macroalgal cover typically suppresses ecosystem services and leads to degraded reef systems. Consequently, policy makers and local coral reef managers should reassess the focus on herbivorous fish protection and consider complementary measures such as watershed management in order to arrest this trend.

## INTRODUCTION

Caribbean coral reefs have experienced major declines over recent decades, with substantial reductions in live coral cover accompanied by concomitant losses in reef accretion and structural complexity (*Schutte, Selig & Bruno, 2010*; *Alvarez-Filip et al., 2011*; *Perry et al., 2015*). Although a wide array of factors have contributed to reef

deterioration including coral diseases, coastal development and climate change, the loss of key herbivores is thought to be a leading driver of ecosystem transition towards macroalgal domination at many reef sites in the region (*Hughes, 1994*; *Jackson et al., 2014*). Macroalgae compete with corals, reducing coral fecundity, recruitment and survival via various mechanisms including overgrowth, shading and allelopathy (*McCook, Jompa & Diaz-Pulido, 2001*; *Hughes et al., 2007*; *Bruno et al., 2009*; *Rasher et al., 2011*). Today, populations of key herbivore taxa are diminished on many Caribbean reefs. The sea urchin *Diadema antillarum* was previously an important grazer in the Caribbean (*Jackson et al., 2001*). In 1983/4 *Diadema* suffered mass mortality across the Caribbean due to putative disease and populations have subsequently shown only limited recovery (*Lessios, Robertson & Cubit, 1984*; *Kramer, 2003*; *Hughes et al., 2010*). Furthermore, long-term overfishing has resulted in marked reductions in herbivorous fish populations at many sites across the region (*Jackson et al., 2001*; *Paddack et al., 2009*).

Given the pivotal role of herbivores in controlling macroalgal growth (*Mumby et al., 2006*), it is widely accepted that restoring populations of key herbivores enhances reef resilience by controlling algal communities and facilitating coral recovery by freeing space for coral recruits (*Nyström, Folke & Moberg, 2000*; *McCook, Jompa & Diaz-Pulido, 2001*; *McManus & Polsenberg, 2004*; *Bruno et al., 2009*). Consequently, coral reefs with high herbivore abundance are expected to have lower macroalgal cover and greater coral cover (*Jackson et al., 2014*; *Kramer et al., 2015*). This paradigm has encouraged global awareness campaigns promoting conservation and fisheries management strategies to protect and restore populations of key herbivorous fishes, particularly parrotfishes (*Jackson et al., 2014*). In the Mesoamerican region, for example, Belize and Guatemala have banned the capture and possession of herbivorous fishes (*Kramer et al., 2015*).

Direct evidence of herbivores' ability to facilitate the maintenance and recovery of resilient coral reefs is limited. Experimental herbivore exclusion studies demonstrate the action of *Diadema* and herbivorous fish grazing on macroalgal cover, although evidence for the impact on corals is limited by the short-term nature and restricted spatial extent of the experiments (*Lirman, 2001*; *Burkepile & Hay, 2006*; *Burkepile & Hay, 2009*; *Hughes et al., 2007*). Observational studies tend to focus on inter-site comparisons without an explicit temporal dimension, rather than tracking long-term reef change trajectories to provide a more in-depth understanding of drivers of ecosystem dynamics (*Karr et al., 2015*). Little consensus exists between studies, which exhibit contrasting patterns between herbivorous fish populations and macroalgal cover. In a Caribbean-wide point-in-time study, *Newman et al. (2006)* found a significant negative correlation between herbivorous fish biomass and fleshy algal biomass, whereas *Loh et al. (2015)* observed that overfished Caribbean sites had lower macroalgal cover than protected sites. For the Northern Mesoamerican Reef of Mexico, *Bozec et al. (2008)* did not observe a relationship between herbivore biomass and macroalgal cover. In a long-term study, *Ilves et al. (2011)* observed increases in both herbivorous fish abundance and algal cover in the Bahamas. On the Northern Florida Reef Tract, *Lirman & Biber (2000)* observed no correlation between algal biomass and cover and fish grazer abundance and consumption rates. *Jackson et al. (2014)* found a significant negative correlation between parrotfish

biomass and macroalgal cover in 16 Caribbean locations; however, no such relationship was observed for a broader data set covering 46 locations. The lack of relationship between herbivorous fish and macroalgal cover is evident for other regions: *Carassou et al. (2013)*, for example, found that macroalgal cover was not correlated with the biomass, density and diversity of macroalgae feeders in the South Pacific.

To further understand the relationship between herbivory pressure and changes in macroalgal cover we propose a simple conceptual framework (Fig. 1). Here, reefs may experience one of four scenarios of temporal changes in fleshy macroalgal cover and herbivorous fish biomass, a widely used proxy for herbivory intensity (*Graham et al., 2015*). Principal ecological drivers are presented for each idealised scenario, although in reality a number of drivers act in conjunction to varying extents. A phase shift from coral to algae domination due to herbivore loss is represented by the scenario in the upper-left quadrant. Here, decreasing herbivory leads to increasing macroalgal cover. Conversely, in the bottom-right quadrant, *increases* in herbivorous fishes result in reduced macroalgal cover. This quadrant represents the scenario sought by management measures and fisheries regulations restricting extraction, particularly of herbivorous fishes (*Halpern, 2003*; *Lester et al., 2009*; *Selig & Bruno, 2010*; *Guarderas, Hacker & Lubchenco, 2011*).

Alternatively, a *positive* relationship may exist between macroalgal cover and herbivorous fish biomass, as represented by the scenarios of the upper-right and bottom-left quadrants of Fig. 1. This may occur when herbivores are food limited, as evidenced by increases in herbivore abundance and biomass following algal growth and by resource competition between *Diadema* and herbivorous fishes (*Hay & Taylor, 1985*; *Carpenter, 1990*; *Adam et al., 2011*). In these scenarios, predominantly external drivers such as nutrient availability, temperature and solar irradiance determine macroalgal cover and herbivorous fish biomass responds according to food availability (*Burkepile & Hay, 2006*; *Ferrari et al., 2012*). Numerous experimental manipulation studies have reported the significant positive impact of nutrient enhancement on primary producer abundance, although herbivory has generally been found to play a greater role (*Burkepile & Hay, 2006*). Contrastingly, few studies have addressed the importance of macroalgal productivity potential relating to environmental factors such as light availability and temperature (*Steneck & Dethier, 1994*; *Ferrari et al., 2012*).

Herbivore and algal community composition also play an important role in herbivore-algal dynamics. Subsequent to the *Diadema* mass mortality event of the early 1980's, herbivorous fishes of the Scaridae and Acanthuridae families are recognised as the primary herbivores on many Caribbean reefs (*Jackson et al., 2014*; *Adam et al., 2015a*). While common *Acanthurus* surgeonfishes have a broad diet feeding on a combination of turf algae, macroalgae and detritus, *Sparisoma* and *Scarus* parrotfishes are more selective (*Burkepile & Hay, 2011*; *Adam et al., 2015a*). *Sparisoma* parrotfishes, with the exception of the excavating *S. viride*, are macroalgae browsers, while *Scarus* spp. primarily graze algal turfs (*Bonaldo, Hoey & Bellwood, 2014*; *Adam et al., 2015b*). Consequently, a suitable mix of herbivores are required in order to both graze turf algae to facilitate coral recruitment and to crop down macroalgal stands to reduce competition with adult coral colonies

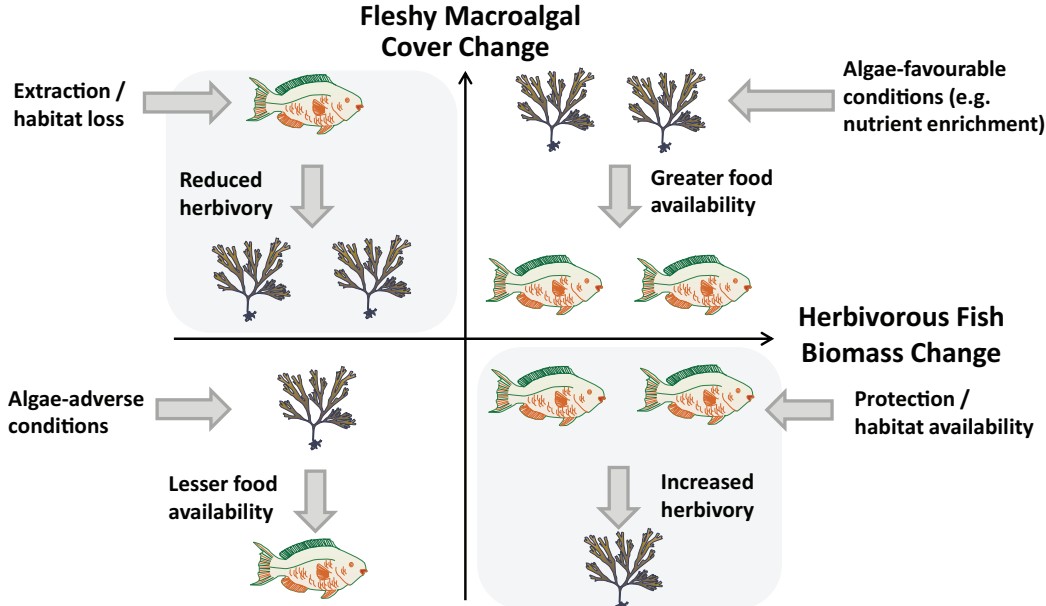

**Figure 1** **Relationship between changes in herbivorous fish biomass and benthic fleshy macroalgal cover.** Possible cause-and-effect scenarios with external drivers are postulated for each quadrant. Fish and algae graphics by Diana Kleine and Tracey Saxby (IAN Image Library, Integration and Application Network, University of Maryland Center for Environmental Science, http://ian.umces.edu/imagelibrary).

(*McCook, Jompa & Diaz-Pulido, 2001*; *Hughes et al., 2007*; *Burkepile & Hay, 2008*). However, herbivores' ability to effectively moderate macroalgal cover is mediated by macroalgal predation defences (*Rasher, Hoey & Hay, 2013*). Such defences are species specific and include morphological, structural, mineral and chemical traits that deter herbivores, with several genera (e.g. *Lobophora*, *Peyssonnelia* and *Codium*) being unpalatable (*Hay, 1997*; *Smith, Hunter & Smith, 2010*). These defences likely influence herbivore feeding preferences and conversely algal community structure is often influenced by herbivore mix, resulting in a complex interaction between the two communities (*Adam et al., 2015a*).

Here, by following individual site trajectories, we examine the prevalence of the four herbivorous fish and macroalgae change scenarios across 85 sites surveyed from 2005 to 2014 along the Mesoamerican Reef. We also consider herbivore functional group composition and trajectories, and compare these with overall trends. Subsequently, we evaluate the potential effects of herbivorous fish biomass, fleshy macroalgal cover and other factors such as degree of protection, on changes in coral cover during the same timeframe. Our hypothesis is that for sites where herbivory increased, fleshy macroalgal cover decreased, and that herbivore biomass and the decline in macroalgal cover are among the main factors explaining coral cover on today's reefs.

## MATERIALS AND METHODS

We used data produced by the Healthy Reefs Initiative (HRI) and the Atlantic and Gulf Rapid Reef Assessment (AGRRA) programs, which include ecological censuses for

398 sites along the Mesoamerican Reef in Mexico, Belize, Guatemala and Honduras from 2005 to 2014. Site selection was based on benthic habitat maps produced by the Millennium Reef Mapping Program, with 200 × 200 m sites randomly selected following stratification by geomorphological characteristics and depth (*Andréfouët et al., 2003*; *Kramer, 2003*). The database contains 85 long-term monitoring sites that were surveyed in 2005/2006 and 2013/2014 over a 7, 8 or 9-year period, a timeframe sufficient to observe ecologically meaningful changes (*Babcock et al., 2010*). Of these sites, 43 were repeatedly surveyed in four time periods (2005/2006, 2009/2010, 2011/2012 and 2013/2014). Sites were located primarily on the fore reef and reef crest at a mean (± Standard Error s.e.m.) depth of 6.9 ± 0.2 m.

Benthic cover and reef fish surveys were performed according to AGRRA protocol, with transects located haphazardly, parallel to the coast (*Lang et al., 2010*). The majority of sites were surveyed at similar times during the summer year-on-year in order to minimise seasonal effects. At each site an average of five to six 10 m-transects were surveyed using point intercept methodology to determine benthic cover including hard coral percentage cover and fleshy macroalgal percentage cover. The abundance and total length (TL) of 81 key reef fish species, including herbivorous fishes of the Scaridae and Acanthuridae families, was recorded in ten 30 m-long, 2 m-wide transects. Reef fish abundance was subsequently converted to biomass density using standard allometric length-weight conversions.

The data analyses focussed on the relation between three ecological indicators for each reef site: herbivorous fish (Scaridae and Acanthuridae) biomass, fleshy macroalgal (excluding turf and calcareous algae) cover and hard coral (scleractinians and *Millepora* spp.) cover. Very few *Diadema* spp. were observed and therefore we focussed on reef fishes as the principal herbivores. For all three ecological indicators, a number of metrics were calculated to evaluate and examine temporal trends: absolute annual change, annual relative rate of change and geometric rate of change. The metrics for each ecological indicator ($I$) were determined as follows:

$$I_{\text{Absolute Annual Change}} = \frac{(I_{t_f} - I_{t_0})}{\Delta t} \tag{1}$$

$$I_{\text{Annual Relative Rate of Change}} = \frac{(I_{t_f} - I_{t_0})}{I_{t_0} \times \Delta t} \tag{2}$$

$$I_{\text{Annual Geometric Rate of Change}} = \left(\frac{I_{t_f}}{I_{t_0}}\right)^{\frac{1}{\Delta t}} - 1 \tag{3}$$

where $I_{t_f}$ is the value of the ecological indicator at the end of the period, $I_{t_0}$ is the initial value and $\Delta t$ is the length of the period (in years). The former two metrics provide complementary information, for example: if an ecological indicator such as coral cover increases from 10–15%, the absolute change Eq. (1) is 5%, while the relative rate of change Eq. (2) indicates that coral cover has increased by 50% relative to its initial value.

Geometric rate of change Eq. (3) was utilised in order to assess and compensate for non-linearity in the relative rate of change, while still providing an interpretable value (*Côté et al., 2005*).

Univariate comparison of ecological indicators was performed using ANOVA, t-tests or non-parametric equivalents (Mann-Whitney U or Wilcoxon Signed Rank tests), based on an assessment of normality and homogeneity of variance using Shapiro-Wilk and Levene tests. To test our first hypothesis, herbivorous fish biomass was compared with fleshy macroalgal cover using Spearman rank-order correlation due to non-normality. Herbivorous fishes were further categorised according to feeding preferences as macroalgae browsers (*Sparisoma* spp., with the exception of *S. viride*), turf grazers/scrapers (*Scarus* spp. and *Acanthurus* spp.) or bioeroders (*Sparisoma viride*) (*Bellwood et al., 2004*; *Burkepile & Hay, 2011*; *Bonaldo, Hoey & Bellwood, 2014*; *Adam et al., 2015a*; *Adam et al., 2015b*). Change in functional group biomass was compared with overall change in herbivorous fish biomass using Spearman rank-order correlation. Furthermore, change in macroalgal cover was compared with absolute levels of overall herbivorous fish and macroalgae browser biomass both graphically by categorising sites by initial fish biomass (based on deciles) and by using Spearman rank-order correlation.

To test our second hypothesis, change in absolute coral cover from 2005/6 to 2013/4 for long-term monitoring sites was modelled using multiple linear regressions as model assumptions were satisfied. To address the common problem of spatial autocorrelation in multi-site studies we performed a Moran's I test on coral cover change by site location which reported no spatial autocorrelation present (Moran's I = 0.070, P = 0.08). The optimum regression model was selected based on Akaike Information Criterion (AIC). Candidate independent variables were selected based on ecological relevance and data availability (Table S1). Potential collinearity among predictor variables was examined using Pearson correlations and variance inflation factors, and outliers were removed on the basis of Cook's D. All statistical analyses were performed using R (*R Core Team, 2014*).

## RESULTS

Here we present herbivorous fish biomass and fleshy macroalgal cover average trends for repeatedly surveyed sites and assess changes in these variables for long-term monitoring sites. Subsequently we examine herbivorous fish feeding guilds and geographic trends for long-term monitoring sites, and assess the effect of protection on site trajectories. Finally we present the ecological drivers of long-term coral cover change.

### Herbivorous fish biomass and macroalgal cover trends

During the time period 2005–2014, regional averages showed a clear trend of increasing fleshy macroalgal cover on the Mesoamerican Reef, while herbivorous fish biomass remained relatively constant. Across 43 sites surveyed repeatedly in four time periods (Fig. 2), mean herbivorous fish biomass did not change significantly (Wilcoxon Signed Rank, Z = 0, P = 1), while mean macroalgal cover doubled during the same period (Wilcoxon Signed Rank, Z = −5.02, P < 0.001). Between 2005/2006 and 2009/2010 mean herbivorous fish biomass decreased and mean fleshy macroalgal cover increased

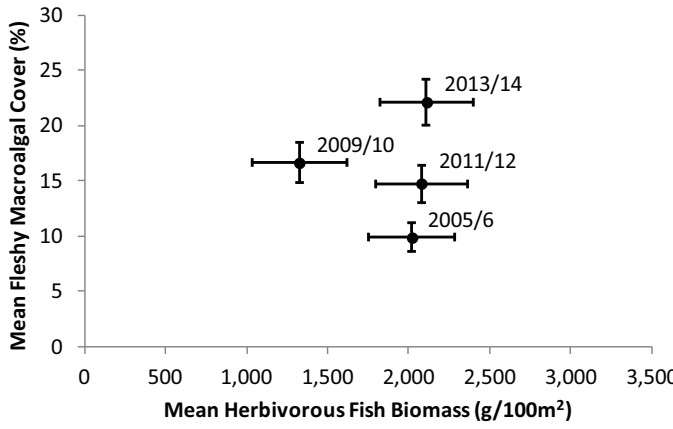

**Figure 2 Temporal trend in mean herbivorous fish biomass and benthic fleshy macroalgal cover on the Mesoamerican Reef.** Mean (± s.e.m.) values are shown for all 43 sites surveyed repeatedly in each monitoring period (2005/2006, 2009/2010, 2011/2012 and 2013/2014). Similar trends were observed for all sites surveyed in consecutive monitoring periods (Fig. S1).

significantly (Wilcoxon Signed Rank, $Z = 3.36$, $P < 0.001$ and $Z = -3.86$, $P < 0.001$, respectively). From 2009/10 to 2011/12 the trend appeared to be reversed, although the changes were not significant for macroalgae (Wilcoxon Signed Rank, $Z = -2.95$, $P = 0.003$ and $Z = 0.59$, $P = 0.55$, respectively; Fig. 2). From 2011/2012 to 2013/2014 macroalgal cover increased significantly, while herbivorous fish biomass remained unchanged (Wilcoxon Signed Rank, $Z = -3.81$, $P < 0.001$ and $Z = -0.35$, $P = 0.73$, respectively; Fig. 2).

Tracking individual trajectories of the 85 long-term monitoring sites surveyed over a 7, 8 or 9-year period permitted a more detailed investigation of the relation between the temporal changes in herbivorous fish biomass and fleshy macroalgal cover. Herbivorous fish biomass ranged from approximately 50–14,000 g/100 m$^2$ and fleshy macroalgal cover ranged from 0–57.5%. There was no correlation between the changes in herbivorous fish biomass and fleshy macroalgal cover for long-term monitoring sites (Spearman, $r_s = -0.11$, $P = 0.35$). Only 7% of sites exhibited increased herbivorous fish biomass and decreased macroalgal cover; 35% of sites displayed decreases in fish biomass and increases in macroalgal cover; almost half of the sites (48%) exhibited increases in both herbivorous fish biomass and macroalgal cover; and 10% displayed decreased fish biomass and macroalgal cover (Fig. 3). Across all sites macroalgal cover increased irrespective of initial conditions of herbivorous fish biomass (Spearman, $r_s = -0.12$, $P = 0.3$; Fig. 4A) and macroalgae browser biomass (Spearman, $r_s = -0.21$, $P = 0.3$; Fig. 4B).

Considering herbivorous fish feeding preferences based on *Bellwood et al. (2004)*, communities of the Mesoamerican Reef present a mixture of guilds with 24.3% macroalgae browsers by biomass in 2013/14 (19.4% in 2005/6), 48.4% (57.3%) turf grazers/scrapers and 27.3% (23.3%) bioeroders. Herbivorous fish biomass and macroalgal cover change were broadly similar between macroalgae browsers and overall results (Fig. 3). Macroalgae browser biomass displayed a slightly greater tendency for increase than overall herbivorous fish biomass, as observed for 61% of sites compared with 55%, and site-level changes in these were correlated (Spearman, $r_s = 0.70$, $P < 0.001$).

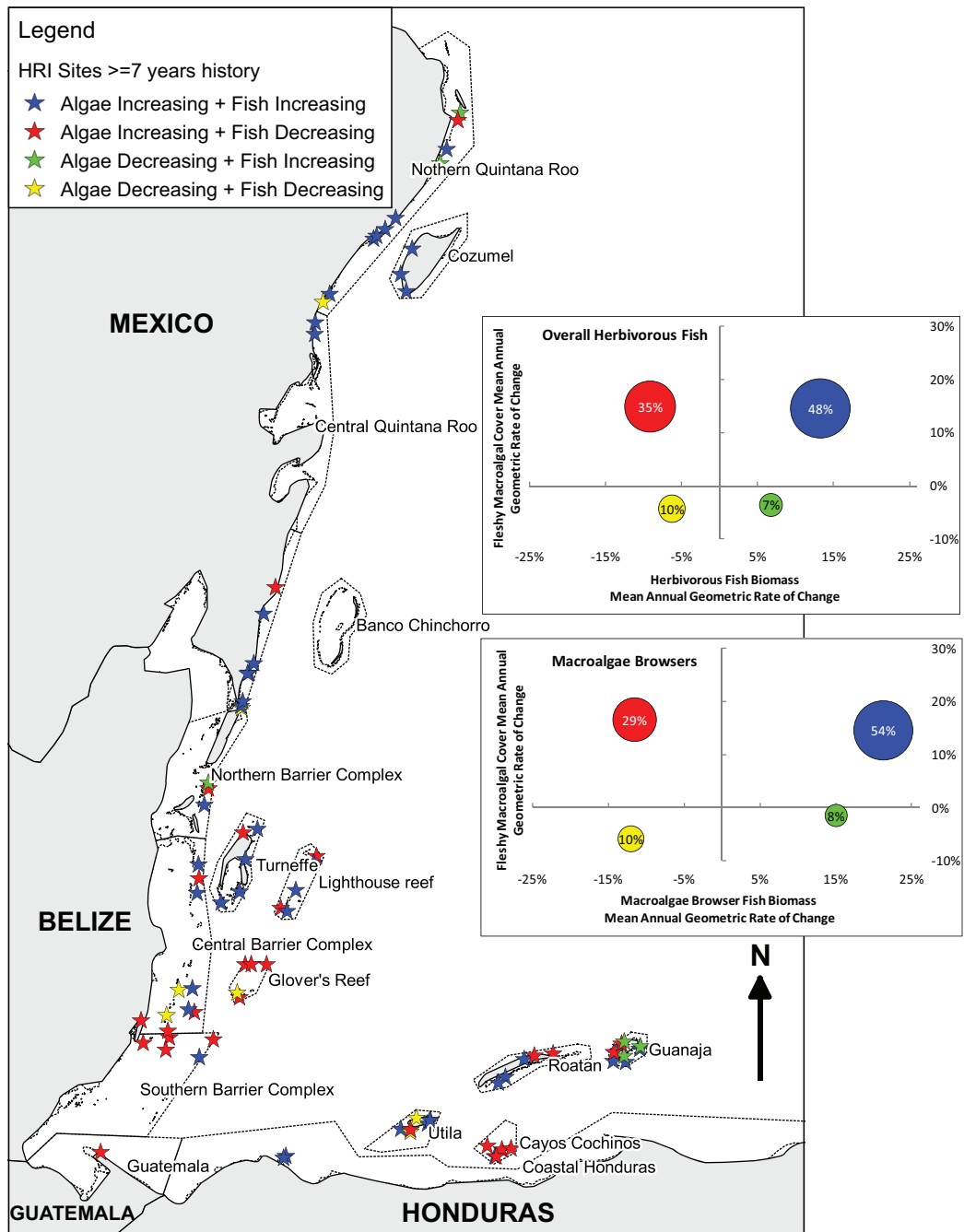

**Figure 3 Long-term herbivorous fish and benthic fleshy macroalgal cover trends on the Mesoamerican Reef.** Map and graphs indicating relationship between changes in overall and macroalgae-browsing herbivorous fish biomass and fleshy macroalgal cover from first (2005 or 2006) to last (2013 or 2014) year for all (85) long-term monitoring sites with ≥ 7 years' history. Map indicates Healthy Reef Initiative regions within countries and locates sites by the relationship between changes in herbivorous fish biomass and fleshy macroalgal cover (*Kramer et al., 2015*). Inset graphs separately indicate the relationship between changes in herbivorous fish biomass and fleshy macroalgal cover, and macroalgae-browsing herbivorous fish biomass and fleshy macroalgal cover. For inset graphs, each circle represents the sites for that quadrant and circle position reflects mean site-level annual geometric rates of change. Circle area represents proportion of sites in that quadrant (also labelled). All (85) sites with ≥ 7 years' of history are plotted in order to provide long-term trends, although the equivalent analysis for sites with ≥ 8 years' of history produced similar results (Fig. S2).

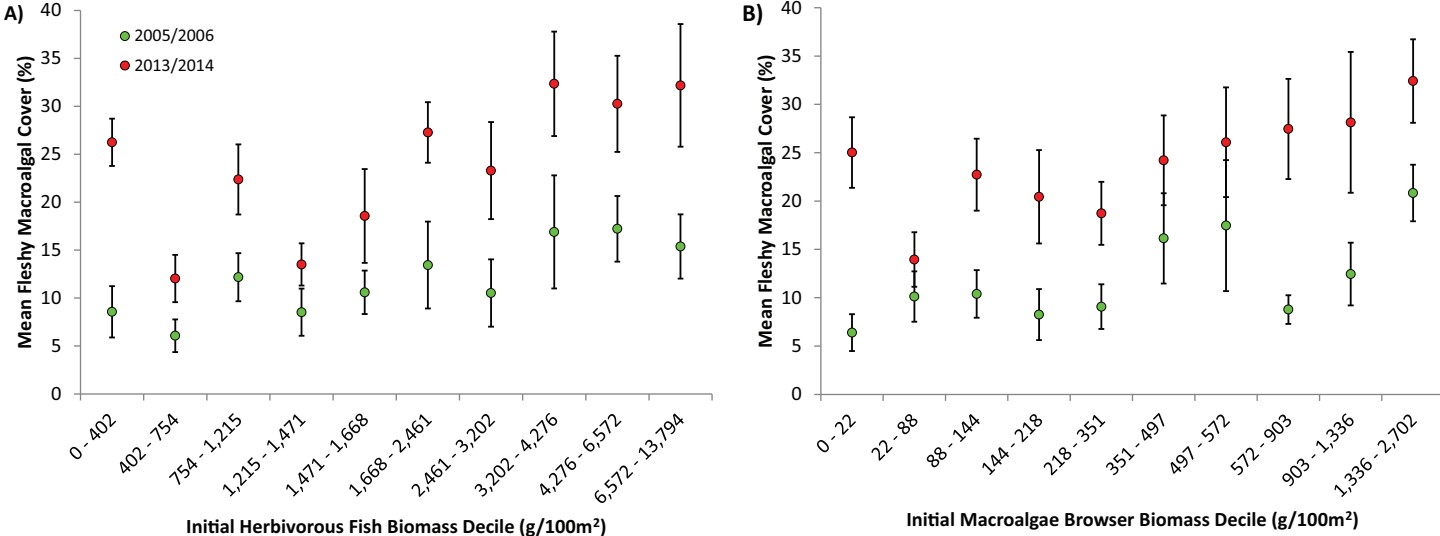

**Figure 4 Effect of initial herbivorous fish biomass on fleshy macroalgal cover on the Mesoamerican Reef.** (A) Mean (± s.e.m.) benthic macroalgal cover in 2005/6 (green symbols) and 2013/14 (red symbols) by initial level of overall herbivorous fish biomass, for all (85) long-term monitoring sites. Sites divided into 10 categories based on initial overall herbivorous fish biomass deciles. (B) Mean (± s.e.m.) benthic macroalgal cover in 2005/6 (green symbols) and 2013/14 (red symbols) by initial macroalgae browser biomass, for all (85) long-term monitoring sites. Sites divided into 10 categories based on initial macroalgae browser biomass deciles.

Geographically, the principal trend was for increasing fleshy macroalgal cover and herbivorous fish biomass in Mexico and northern Belize, including the atolls of Turneffe and Lighthouse Reef, but for increasing fleshy macroalgal cover and decreasing herbivorous fish biomass to the south in south-central and southern Belize, Glover's Reef, Guatemala and Cayos Cochinos, Honduras (Fig. 3). However, the Bay Islands of Honduras were exceptions to this broad north-to-south trend with Guanaja island displaying the highest proportion of sites with increasing herbivorous fish biomass and decreasing macroalgal cover. The only other three sites that experienced increasing herbivorous fish biomass and decreasing macroalgal cover were located at Isla Mujeres and Puerto Morelos in Mexico, and San Pedro in Belize.

Of the 85 long-term monitoring sites, 12 sites were located within No Take Zones (NTZs) where all extractive practices are prohibited, 47 were within Marine Protected Areas (MPAs) but not NTZs where reefs benefit from regulation but some extractive practices are permitted, and the remaining 26 were unprotected. The level of protection was observed to affect the initial levels of fleshy macroalgal cover and herbivorous fish biomass, in addition to changes in these over time. In 2005/6, sites within NTZs exhibited similar herbivorous fish biomass and fleshy macroalgal cover to sites located elsewhere within MPAs (Mann-Whitney, U = 197, Z = −1.60, P = 0.11; and U = 297, Z = 0.28, P = 0.78 respectively; Fig. 5). Protected sites (both MPAs and NTZs) displayed significantly higher initial macroalgal cover than unprotected sites (Mann-Whitney, both U ≥ 247, Z ≥ 2.85, P ≤ 0.003; Fig. 5), but only protected sites outside of NTZs exhibited significantly higher initial herbivorous fish biomass than unprotected sites (Mann-Whitney, MPA vs unprotected, U = 819, Z = 2.40, P = 0.016; NTZ vs unprotected, U = 161, Z = 0.16, P = 0.89; Fig. 5). Along the protection gradient (from unprotected, through MPA to no-take

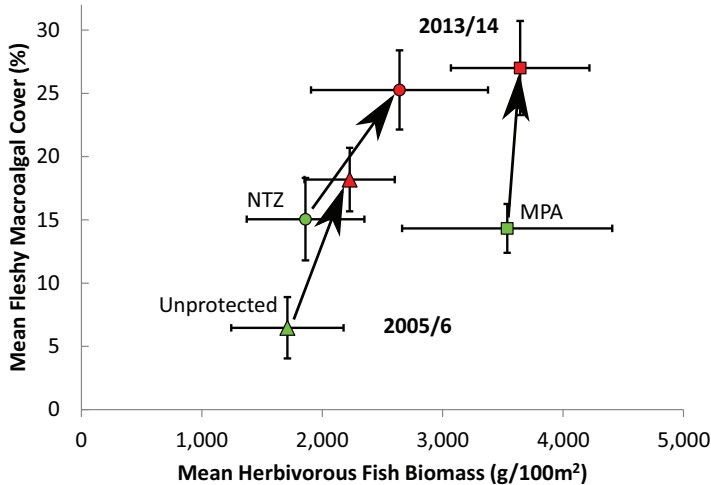

**Figure 5 Effect of protection on herbivorous fish biomass and fleshy macroalgal cover on the Mesoamerican Reef.** Mean (± s.e.m.) herbivorous fish biomass and benthic macroalgal cover in 2005/6 (green symbols) and 2013/14 (red symbols) by level of protection, for all (85) long-term monitoring sites. Unprotected = sites outside Marine Protected Areas (n = 26), MPA = sites inside Marine Protected Areas but not within No Take Zones (n = 47), NTZ = sites inside No Take Zones within Marine Protected Areas (n = 12).

protection), sites appeared to experience a greater increase in herbivorous fish biomass and a lesser increase in macroalgal cover, although the differences were not statistically significant (ANOVA, annual geometric rate of change in fish biomass, $F_{2,82} = 0.04$, $P = 0.97$; annual geometric rate of change in macroalgal cover, $F_{2,80} = 1.01$, $P = 0.37$).

## Predicting coral cover change

Across all 85 long-term monitoring sites, mean (± s.e.m.) hard coral cover increased significantly from 12.2 ± 0.8% in 2005/6 to 15.0 ± 0.8% in 2013/14 (Wilcoxon Signed Rank, $Z = -3.81$, $P < 0.001$). Individual sites displayed varying trajectories with annual changes in coral cover ranging from −3.1 to +2.7%. The optimum linear regression model for the annual absolute change in hard coral cover displayed a modest but significant fit (Adjusted $R^2 = 0.18$, $F_{7,74} = 3.57$, $P = 0.002$). The model included seven predictor variables (Table S1), of which four were significant: MPA, country (Honduras), annual logarithmic change in herbivorous fish biomass and initial hard coral cover (Fig. 6). Interpreting these significant variables, sites within MPAs experienced greater increases in coral cover than unprotected sites; Honduran sites experienced lesser increases in coral cover than other countries; and increases in herbivorous fish biomass corresponded with increases in coral cover. Initial coral cover was the most significant predictor of coral cover change and therefore in order to further understand its impact, sites were categorised as low (< 10%), medium (10–20%) or high (≥ 20%) initial coral cover. For the 12 long-term monitoring sites with high initial coral cover, mean coral cover decreased over time (mean ± s.e.m. annual change in coral cover of −0.6 ± 0.4%), while coral cover increased for sites with lower levels of initial cover (low initial coral cover, +0.6 ± 0.1%; medium initial coral cover, +0.4 ± 0.2%).

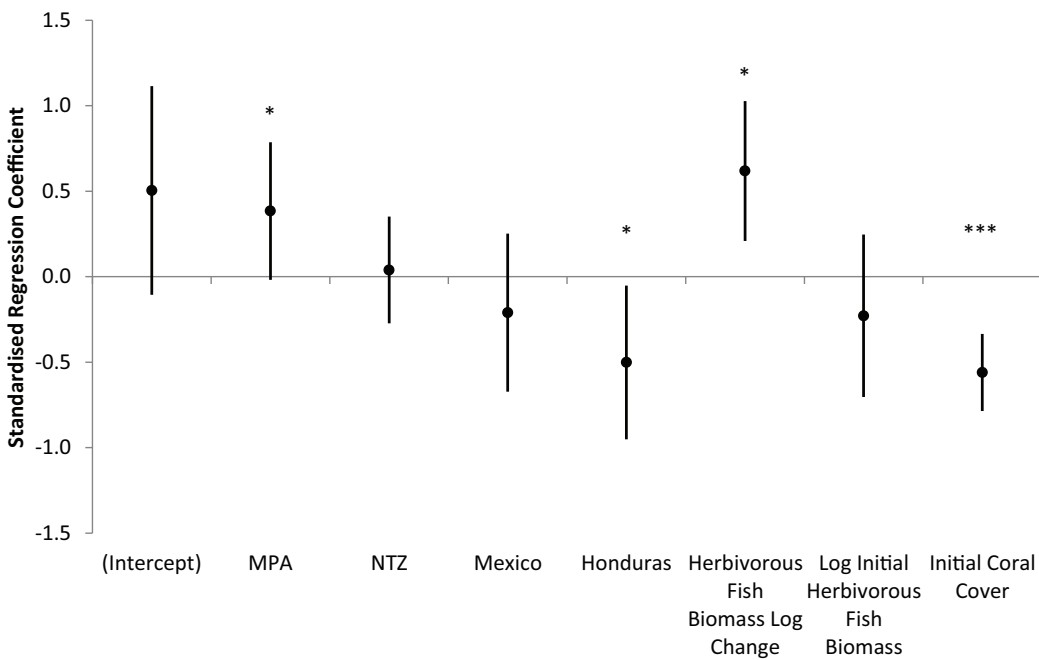

**Figure 6 Prediction of coral cover change on the Mesoamerican Reef.** Standardised regression coefficients for independent variables in AIC-selected optimum model of annual absolute change in hard coral cover from 2005/6 to 2013/14 for all long-term monitoring sites. MPA and NTZ are binary indicators of the location of sites within a Marine Protected Area or No Take Zone, respectively. Mexico and Honduras are binary indicators of the location of sites within those countries. Coefficients reflect the number of standard deviations change in the dependent variable for a one standard deviation increase in each independent variable, while controlling for all other independent variables. Error bars are coefficient standard errors. Significant variables (in non-standardised regression) are highlighted (*** 0.001 level, * 0.05 level).

Despite not being selected in the optimum regression model, we further explored the relationship between coral and macroalgal cover due to the long-term ecological shifts reported on many Caribbean reefs. Across all 85 long-term monitoring sites, mean (± s.e.m.) macroalgal cover increased significantly from 12.0 ± 1.1% in 2005/6 to 24.1 ± 1.5% in 2013/14 (Wilcoxon Signed Rank, Z = −7.07, P < 0.001). We observed little or no relationship between coral and macroalgal cover since macroalgal cover consistently increased irrespective of changes in coral cover (Fig. S3). All initial coral cover categories (low (< 10%), medium (10–20%), and high (≥ 20%)) experienced increases in fleshy macroalgal cover, and initial macroalgal cover, similarly categorised, did not impact coral cover changes over time (ANOVA, $F_{2,82}$ = 1.10, P = 0.34).

## DISCUSSION

Substantial changes in the ecological composition of the Mesoamerican Reef were evident in a time span of only nine years. The principal trend is for increasing fleshy macroalgal cover, as observed at 83% of long-term monitoring sites (Fig. 3). Mean absolute cover of fleshy macroalgal cover increased by approximately 12% in the region between 2005 and 2014 (Fig. 2). Mean herbivorous fish biomass remained relatively stable (Fig. 2), although displaying substantial site variation, with 55% of sites showing an increase in
herbivorous fish biomass between 2005 and 2014 (Fig. 3). The scenario of both increasing fleshy macroalgal cover and herbivorous fish biomass was observed at 48% of the 85 sites while the 'desirable' scenario of increasing herbivorous fish biomass and decreasing macroalgal cover was the least frequent of all four scenarios (Fig. 3). Similar trends were observed for the macroalgae-browsing fish guild (Fig. 3), with site-level macroalgae browser biomass change correlating with overall herbivorous fish biomass change. This suggests that fish herbivory was not a major driver of fleshy macroalgal cover change on the majority of surveyed sites across the Mesoamerican Reef (Fig. 3).

The clear pattern of increasing macroalgal cover and stable herbivorous fish biomass on Mesoamerican reefs contrasts with the widely accepted coral reef top-down herbivore control paradigm and management recommendations that advocate increasing herbivory to control fleshy macroalgal cover (*Nyström, Folke & Moberg, 2000*; *McCook, Jompa & Diaz-Pulido, 2001*). This result is consistent with a multi-decadal study reporting that macroalgal cover was not related to long-term parrotfish losses due to fishing in the Philippines (*Russ et al., 2015*). Furthermore, we found that coral cover on the Mesoamerican Reef was low and unrelated to macroalgal cover. Since both coral cover and reduced herbivory were not responsible for increasing macroalgal cover, external factors may have played a role. For the Mesoamerican Reef region a growing body of evidence shows that rising nutrient levels is a worsening problem that may be accelerating macroalgal increase. In the Mexican Caribbean, previous studies have observed elevated nutrient input to coral reefs due to coastal development (*Baker, Rodríguez-Martínez & Fogel, 2013*; *Hernández-Terrones et al., 2015*) and the subsequent degradation of reef systems (*Bozec et al., 2008*). In southern Belize and Honduras, riverine discharge and escalating reef sediment and nutrient loads associated with urban and agricultural run-off may have played a role in increasing macroalgal cover (*Burke & Sugg, 2006*; *Carilli et al., 2009*; *Soto et al., 2009*). Our finding that fish herbivory is not responsible for macroalgal cover trends contrasts the results of herbivore exclusion studies, which emphasize the relative importance of herbivory over nutrient availability (*McClanahan, Cokos & Sala, 2002*; *Burkepile & Hay, 2006*; *Burkepile & Hay, 2009*; *Sotka & Hay, 2009*). However, contrary to the present study, such experiments tend to be conducted on restricted spatial and temporal scales. Unfortunately, site nutrient data are not widely available for the Mesoamerican Reef, impeding a quantitative exploration of this effect in our analyses.

One alternative that could partially explain the rapid increases in fleshy macroalgae across the Mesoamerican Reef is that reef ecosystems passed critical thresholds beyond which herbivorous fishes are unable to control macroalgae due to either excessive algal production and/or insufficient herbivory (*Mumby, Hastings & Edwards, 2007*). This is particularly relevant given that Caribbean reefs may suffer from insufficient herbivory due to both the limited population recovery of *Diadema antillarum* subsequent to previous mass mortality and the inability of herbivorous fish to adequately compensate for this loss (*McClenachan, 2009*; *Paddack et al., 2009*; *Hughes et al., 2010*). However, excessive algal production is unlikely on the Mesoamerican Reef as regional average macroalgal cover increased from only 10% in 2005/6 to 22% in 2013/14 (Fig. 2), values that are likely considerably below ecosystem thresholds for Caribbean reefs (*Bruno et al., 2009*).

Furthermore, an examination of macroalgal change by absolute levels of herbivorous fish biomass revealed increasing fleshy macroalgal cover even for those sites with the highest initial fish biomass (the uppermost deciles possessed average overall herbivorous fish biomass and macroalgae-browsing fish biomass of 9,065 g/100 m$^2$ and 1,762 g/100 m$^2$ respectively; Fig. 4). Although there is little consensus on Caribbean reef herbivorous fish thresholds, a global assessment of the status of coral reef herbivorous fishes identified only 9 of 132 localities as having herbivorous fish biomass greater than 9,000 g/100 m$^2$, suggesting this to be a high benchmark (*Mumby, Hastings & Edwards, 2007*; *Edwards et al., 2014*). In addition, a negative correlation between Caribbean reef herbivorous fish biomass and fleshy algal biomass has been previously observed with a site maximum of only 7,000 g/100 m$^2$ approximately (*Newman et al., 2006*).

The threshold hypothesis would be particularly relevant if the decline in average herbivorous fish biomass between 2005/6 and 2009/10 resulted in changes in the relative proportion of key functional groups, favouring non-macroalgae-browsing species (*Adam et al., 2015b*; Fig. 2). However, the relative proportions of the three main herbivorous fish functional groups remained stable during the study period (Fig. S4). Additionally, a close examination of those sites that suffered the greatest herbivorous fish biomass losses between 2005 and 2009 revealed that these sites experienced similar macroalgal growth from 2009 to 2014 compared with other sites (Fig. S5). This suggests that the observed rapid increases in fleshy macroalgae are not due to Mesoamerican reefs passing critical thresholds of excessive algal production and/or insufficient herbivory.

Fish populations may impact benthic communities indirectly through mediation of benthic competition. Sponges are a major component of Caribbean coral reef benthos that compete for space with corals and macroalgae (*Loh et al., 2015*). Sponges' competitive superiority over corals is well documented and likely due to a number of mechanisms including shading, smothering and allelopathy (*Porter & Targett, 1988*; *Loh et al., 2015*). Overfishing of spongivorous parrotfishes and angelfishes has been shown to alter ecosystem dynamics through the alleviation of predation pressure on sponges (*Loh & Pawlik, 2014*; *Loh et al., 2015*). Therefore, it is likely that at sites with high parrotfish biomass, spongivory will control benthic sponge cover, indirectly benefiting macroalgal and coral communities via reduced benthic competition. Unfortunately we could not further explore the role of sponges in shaping benthic interactions as the survey protocol does not focus on producing reliable sponge cover information (*Lang et al., 2010*).

Coral recovery on the Mesoamerican Reef was related to MPA protection and increasing biomass of herbivorous fish, but not via the expected mechanism of macroalgal declines through fish herbivory. Alternative mechanisms for the effect of protection on reef corals are less well studied, but may include reduced disease prevalence, and diminished physical reef damage through regulation of fishing and recreational diving practices (*Hasler & Ott, 2008*; *Lamb et al., 2015*). Replenished fish communities inside marine reserves can also drive coral recovery through ecological processes not necessarily linked with herbivory. For example, trophically diverse fish communities inside marine reserves have been shown to ameliorate coral disease prevalence, although the pathways through which this takes place remain unclear (*Raymundo et al., 2009*). Alternatively, coral

cover and complexity may influence herbivorous fish populations, rather than vice-versa, or the relationship may be purely correlative with both indicators being driven by marine protection (*Halpern, 2003*; *Selig & Bruno, 2010*; *Alvarez-Filip, Gill & Dulvy, 2011*).

Reef protection has a positive impact on herbivorous fish biomass and coral cover, although fleshy macroalgal cover continued to increase at most sites. Although protection impacted herbivorous fish biomass and macroalgal cover trajectories (Fig. 5), initial differences between protected and unprotected sites tend to persist, with unprotected sites continuing to display lower macroalgal cover. This may be attributable to reserve age, as protected sites were located within reserves designated in 2003 (± 1 year) on average, and studies have shown that protection influence may be subject to a lag effect (*Selig & Bruno, 2010*; *Babcock et al., 2010*). Furthermore, the use of protection categories (No Take Zones (NTZs), MPAs but not NTZs, and unprotected) is a coarse measure of the actual range of protection and fishing pressure experienced at sites. Additionally, local conditions and reserve regulations often obfuscate protection impact due to variability of internal factors such as reef community structure and enforcement level, and external impacts including local stressors and global climate change (*Mora et al., 2006*; *McClanahan et al., 2006*; *Selig, Casey & Bruno, 2012*). Finally, trophic effects may play a role since trophic cascades are expected when populations of large predators are enhanced due to protection (*Estes et al., 2011*). The protection of piscivores, for example, may result in herbivore reduction and consequently elevated macroalgal growth inside marine reserves. However, studies that explored this question have found that changes in predator populations do not discernibly influence or are even positively correlated with the density, size, and biomass of herbivorous fishes, suggesting that top-down forces may not play a strong role in regulating large-bodied herbivorous fish on coral reefs (*Mumby et al., 2006*; *Houk & Musburger, 2013*; *Rizzari, Bergseth & Frisch, 2015*).

## CONCLUSIONS

Despite the long-term reduction of herbivory capacity reported across the Caribbean, the Mesoamerican Reef displayed relatively low macroalgal cover at the onset of this study. Subsequently, during the last decade, fleshy macroalgal cover increased rapidly on Mesoamerican reefs. Herbivorous fish populations were not responsible for this trend, contrasting the coral reef top-down herbivore control paradigm and implicating the role of external factors in making environmental conditions more favourable for algae. Increasing macroalgal cover typically suppresses ecosystem services and leads to degraded reef systems. Consequently, policy makers and local managers should consider complementary protection measures such as watershed management, in addition to herbivorous fish protection, in order to arrest this trend.

## ACKNOWLEDGEMENTS

The authors recognize the invaluable efforts of Healthy Reefs Initiative (HRI) partner organizations and individual field researchers who collaborated over the years in collecting the data. Data contributors are listed in HRI Report Cards and supplemental reports, available at www.healthyreefs.org. In particular Ian Drysdale, Marisol Rueda,

Ana Giro and Roberto Pott are recognised for coordinating field surveys, training and data entry. The HRI database is processed and managed in conjunction with Ken Marks and Judith Lang. Our manuscript was significantly improved by insightful comments from R. Iglesias-Prieto, J. Bruno and one anonymous reviewer.

### Funding

Data collection was funded by the Summit Foundation, Oak Foundation, JRS Biodiversity Foundation and an anonymous donor organization. A.S. was supported by a PhD scholarship (No. 667112/587102) from CONACyT and an RSG grant from The Rufford Foundation. L.A.-F. was funded by the Mexican Council of Science and Technology (CONACyT, PDC-247104). The funders had no role in study design, data collection and analysis, decision to publish, or preparation of the manuscript.

### Competing Interests

The authors declare that they have no competing interests.

### Author Contributions

- Adam Suchley conceived and designed the experiments, analyzed the data, wrote the paper, prepared figures and/or tables, reviewed drafts of the paper.
- Melanie D. McField wrote the paper, reviewed drafts of the paper, oversaw data collection.
- Lorenzo Alvarez-Filip conceived and designed the experiments, wrote the paper, reviewed drafts of the paper.

### Data Deposition

The data used for this study is available in the Healthy Reefs Initiative database accessible online through http://data.healthyreefs.org. Registration is required by Healthy Reefs to access the data.

### Supplemental Information

Supplemental information for this article can be found online at http://dx.doi.org/10.7717/peerj.2084#supplemental-information.

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
