# Peer review of "Rapidly increasing macroalgal cover not related to herbivorous fishes on Mesoamerican reefs"

_PeerJ, doi:10.7717/peerj.2084_

## Round 0.1 · original submission · Major Revisions

The authors have done an important and timely study, as both reviewers indicate. I agree, however, that this manuscript requires some revisions before publication. Once properly revised, I suspect that this paper will be widely cited.

Please consider the reviewers' comments carefully and provide point-by-point responses to them. Both reviewers commented on the lack of multivariate analyses of the data, and these should be undertaken, even if reported in addition to the univariate analyses.

The authors are missing important references to recent work that supports their own, including the Loh et al PeerJ paper, which also finds no relationship between fish abundance and seaweed cover, and the recent work of Burkepile et al, which suggests that higher fish abundance results in greater nutrient inputs that enhance seaweed cover.

Both reviewers indicate that the terrestrial pollution angle is unsupported by the data and there is little evidence that terrestrial nutrients make it to the reef -- these speculations should be removed, as there are better explanations for the data.

Reviewer 2 has specific comments about MPAs that need to be addressed.

The title is vague and does not convey the importance of the paper -- consider something like "Parrotfish restoration does not reduce macroalgae on the Mesoamerican coral reef" or "No reduction of macroalgal cover with increasing parrotfish abundance on the Caribbean mesoamerican reef."

·

Basic reporting

All basic reporting criteria have been met.

Experimental design

The experimental design is very good. The authors have excellent spatial relocation (many sites), across a huge geographic scale, as well as temporally repeated near-annual surveys for a decade. The survey methods are fine. The statistical analysis seems fine (although I was a little surprised by the use of Pearson correlation for the algal data). The study was conducted at a very high technical level and the methods are described in detail. Moreover, the code and data will be made available. The study has a clear purpose and hypothesis and the data collected match the study question.

Validity of the findings

I believe the data are robust and that the main inferences (regarding algae and grazers) are valid and supported by the data/analyses. The only inference I question (see below) is the purported affect of herbivores (biomass) on corals (cover). In general, I think the authors would be wise to be very careful about inferring causal direct relationships here, e.g., when two variables are related, inferring an effect of one on the other. I very much support this type of macro ecological approach, but one does have to be prudent in interpreting such data.

Additional comments

I think this is an excellent, well-executed, high impact and novel study / manuscript. I wouldn’t have been surprised to have gotten this to review at PNAS, Ecology or similar. (I suspect the authors had a hard time in review elsewhere given the controversial nature of the findings - the authors data contradicts a paradigm of coral reef ecology and conservation.

I made several minor editorial suggestions on the Word manuscript file, which I will upload. These will require only minor revision.

My two main suggestions (which are related) for changes I think would improve the manuscript are:

1) Edit the end of the Abstract and Discussion (the conclusions section) to focus more on the algae-parrotfish results and their implications for reef restoration, management and “resilience”. Both sections drift somewhat and could be much more forceful and point out these results contradict a major paradigm in the field and should cause us to question the advice we scientists have been giving to policy makers and managers. Also, both sections spend far too much time on the putative relationship between coral cover and parrotfish biomass (which I think is spurious, unclear, and not a major funding).

2) I am dubious about the inferences made from the coral cover change data, particularly about the purported role of parrotfishes in facilitating the observed trends. First, the mean regional increase in coral cover is trivial: ~3%, which is barely detectable, even with benthic videos (i.e., the change is below the precision of the method). Second, it is ecologically meaningless. Third, it could be an increase of weedy taxa, and not a positive management outcome. Fourth, given several factors were related to coral cover change, I don’t think the authors can assume that each had a direct, causal effect on coral cover (be it positive or negative). I don’t know what mechanism could plausibly explain how an increase in grazer biomass could facilitate an increase in coral cover (ruling out a role of grazing). In fact, I think it is more likely that living coral attracted parrotfishes (e.g., check this video out: https://vimeo.com/105031573) or that they both benefitted (slightly) from MPAs and other local management activities (i.e., their spatiotemporal dynamics could be related, but not causally). Finally, there are no graphics depicting the coral cover trend data or how they related to the covariates used in the model. Given the slight change in coral, I would be very careful in not over interpreting the results. And I’d cut much or all of the discussion about causes of the minor coral trend from the Abstract and Conclusions - both of which almost focus on this minor and tricky result. Focus instead on the macroalgal-herbivore result, which is much clearer, interpretable, and important.

Also:

What about coral loss? Couldn’t that possibly explain macroalgal increase? i.e., by increasing the availability of an important resource (space) and also by increasing primary production, thus reducing relative grazing pressure. At least explain why not and discuss the idea.

The Introduction is fantastic. A super, clear, synthetic summary of the relevant science. The literature reviewed between lines 68 and 93 is eye-opening. I knew of some of these studies, and know of others not cited, but seeing them all together like this makes a powerful argument. I really like the introduction of the conceptual model too.

The Discussion section is also great. I appreciate the care in which the authors considered alternative hypotheses and related their findings to other work. I would only cut/edit the text on the change in coral cover and better explain the text about thresholds (presumably of grazing, primary production, and the ratio between the two).

The observed increase in macroalgal cover is pretty amazing (and somewhat surprising), but is concordant with data my lab has collected in Belize.

Reviewer 2 ·

Basic reporting

In general, it’s good to see more work, especially over a large spatio-temporal scale, add nuance to the herbivore-macroalga-coral paradigm, which tends to be overly simplistic and limited to specific locations. Saving Caribbean reefs will take much more than preserving populations of parrotfishes. The writing overall is fine; some compound adjectives need hyphenation and a couple of statements could be phrased more clearly. These are highlighted in the general comments.

In terms of background information, I suggest providing some context for readers who are not familiar with the history and ecology of Caribbean coral reefs. E.g. stating the levels of coral decline (line 37), explaining the Diadema story (that it is a sea urchin, mass mortality due to putative disease) a little more (lines 48-50), and listing the “key herbivorous fishes” (line 62).

A major omission is the lack of discussion of the role of sponges in coral reef benthic communities in the Caribbean, especially when the research question assessed spatial competition between hard corals and macroalgae. Sponges are one of the dominant benthos and spatial competitors in the region, and are a food source for parrotfishes as well (Loh & Pawlik, 2014, PNAS). The main competition for space among reef benthos is likely between macroalgae and sponges, and the consideration of sponges in the analysis might partially explain why macroalgal cover is high at sites with lots of parrotfishes (because fast-growing sponges are grazed down and don’t compete with macroalgae).

Another explanation for a positive relationship between fish biomass and macroalgal cover is the input of nutrients from fish excretion to the reef (Burkepile et. al. 2013, Scientific Reports). The conclusion would be stronger with a more thorough discussion of the role of Diadema in limiting macroalgal growth before the mass mortality event. The lack of recovery through the study period, and the inability of herbivorous fish to adequately fill this niche in Diadema’s absence probably explain why a strong relationship between herbivorous fish biomass and macroalgae wasn’t observed in this study.

Experimental design

Why were univariate comparisons used for the ecological indicators instead of a multivariate analysis? The variables are not likely independent as they come from the same transects, e.g. macroalgal and hard coral cover. In addition, running parallel tests from a single dataset inflates the probability of Type 1 errors. Was this accounted for? The manuscript will be improved with a multivariate analysis to look at differences among sites and corresponding correlations.

From Line 271, MPAs without no-take zones are a non-standardized factor, without consistent protection from overfishing for all MPAs. MPA regulations could range from being fully open to fisheries exploitation to restricting the fishing of some species, thus designating sites as MPA vs. non-MPA isn’t ideal as a predictive factor. Instead, would it be possible to provide an index of fishing pressure across all sites? E.g. specifically assess fishing regulations and enforcement capacity at each sites.

Validity of the findings

The manuscript lists nutrient inputs from terrestrial run-offs as having major impacts on macroalgal cover, but this hypothesis (and other putative factors) was not explicitly tested. Are there any nutrient data available or previous work that can be cited, even for a subset of sites, to support this assertion? Also, it is possible to calculate the distance from each site to land or the nearest river discharge as a proxy for terrestrial runoff contribution, and this would be a valuable addition to the data analysis, especially with terrestrial runoffs coming up a few times during the discussion.

In addition, the discussion of fishing protection in NTZs and MPAs was limited to herbivorous fishes. I’d like to see a short discussion of the ecological impacts of preserving populations of large piscivores that eat herbivores in well-managed marine reserves, given the authors’ assertions that herbivores benefit corals and that “local marine protection” (Line 462) is an important management tool for healthy coral populations.

Additional comments

Here are line-by-line comments:
Line 76: Hyphenate “Caribbean-wide”
Line 81: The findings for the Northern Florida Reef Tract are not a contrast to the previous statements
Line 86: What is “this pattern”?
Line 98-100: Are you talking about management measures (MPAs) and fisheries regulations specific to protecting herbivorous fish? Be clearer.
Line 124-125: Sparisoma parrotfishes are major sponge predators too!
Line 268: Specify what the “specific trend” is
Line 272: Throughout the manuscript, distinguish “protection” from overfishing from other types of protection, e.g. protection from physical forces
Line 299: The positive correlation between herbivorous fish and hard coral cover might be mediated by predatory limitation of fast-growing, palatable sponges
Line 320: “Biotic” is more appropriate here than “ecological”
Line 330: Hyphenate “macroalgae-browsing” here and for the rest of the paragraph
Line 332: Diadema was probably a much more effective macroalgae grazer
Line 338-340: This statement seems stand-alone. Expand on what it means for your study
Line 344-345: Why those factors specifically? Those were all not tested in your study
Line 348: What do you mean by “reef structural change,” and how is that related to macroalgal cover?
Line 350-352: Are sites at southern Belize and Honduras associated with increasing macroalgae and high fish biomass?
Line 357: Herbivorous fish abundance can also increase if piscivorous fish have already been fished out, in addition to a bottom-up factor
Line 363-366: I know consistent reports of macroalgal species were not available, but is this statement supported by any specific examples in the dataset?
Line 371: The Diadema aspect should come in here
Line 398-400: Yes, agreed! Sponges and macroalgae probably have stronger interactions
Line 427-429: While the authors may not have access to consistent nutrient data, it is possible to map the distance of each site to land or the nearest river discharge for a proxy of terrestrial inputs, especially as the authors argue this is a “substantial” impact. Because of the large variability in MPA regulations and level of enforcement, there may not be much difference between “protected” and “unprotected” sites
Line 455-456: I would say the lack of Diadema recovery is a major factor too
Line 462: “Protection” from overfishing? Was this specifically targeted in your assessment of MPA vs non-MPA sites?

---

## Round 0.2 · Minor Revisions

The authors have done a good job of addressing the concerns of the reviewers. I encourage them to make their response document available for readers -- this is a service that is unique to PeerJ, and useful when readers have questions that are addressed in the supplementary documents. Just a few minor edits are still needed:

The title much better reflects the importance of the contribution, but "fish" should be "fishes" -- and I'm not sure of the directed nature of "fishes being unable to do something" is a good choice. I would ask the authors to consider "No relationship between macroalgal cover and herbivorous fish abundance..." which is more neutral, or something similar.

The summary paragraph beginning line 70 of other broad-scale studies is missing the important reference to Loh et al (2015), which compared fish-trapped sites vs MPAs at 69 sites across the Caribbean with lower mean macroalgal cover at less-fished sites. See Fig. 4C. As indicated in the text, although angels and parrots were counted, this was really a "fish vs. no-fish" comparison, with the surprising result of greater macroalgal cover when fish abundance high.

---

## Round 0.3 · accepted · Accept

Nice job -- this is an important contribution to the literature.